# Effect and Mechanism of Soluble Starch on Bovine Serum Albumin Cold-Set Gel Induced by Microbial Transglutaminase: A Significantly Improved Carrier for Active Substances

**DOI:** 10.3390/foods12234313

**Published:** 2023-11-29

**Authors:** Haoting Shi, Changsheng Ding, Jianglan Yuan

**Affiliations:** Cooperative Innovation Center of Industrial Fermentation (Ministry of Education & Hubei Province), College of Bioengineering and Food, Hubei University of Technology, Wuhan 430068, China; htshi0513@163.com (H.S.); dingchangsheng2023@163.com (C.D.)

**Keywords:** bovine serum albumin, soluble starch, cold-set gel, microbial transglutaminase, glucono-δ-lactone

## Abstract

Soluble starch (SS) could significantly accelerate the process of bovine serum albumin (BSA) cold-set gelation by glucono-δ-lactone (GDL) and microbial transglutaminase (MTGase) coupling inducers, and enhance the mechanical properties. Hardness, WHC, loss modulus (G″) and storage modulus (G′) of the gel increased significantly, along with the addition of SS, and gelation time was also shortened from 41 min (SS free) to 9 min (containing 4.0% SS); the microstructure also became more and more dense. The results from FTIR, fluorescence quenching and circular dichroism (CD) suggested that SS could bind to BSA to form their composites, and the hydrogen bond was probably the dominant force. Moreover, the ability of SS to bind the original free water in BSA gel was relatively strong, thereby indirectly increasing the concentration of BSA and improving the texture properties of the gel. The acceleration of gelling could also be attributed to the fact that SS reduced the negative charge of BSA aggregates and further promoted the rapid formation of the gel. The embedding efficiency (EE) of quercetin in BSA-SS cold-set gel increased from 68.3% (SS free) to 87.45% (containing 4.0% SS), and a controlled-released effect was detected by simulated gastrointestinal digestion tests. The work could put forward new insights into protein gelation accelerated by polysaccharide, and provide a candidate for the structural design of new products in the food and pharmaceutical fields.

## 1. Introduction

Gels have attracted the attention of food researchers, due to good texture properties and a high-moisture network structure in recent years. Some proteins are able to form gel through constructing a three-dimensional network in their systems, and are thereby widely used in the processing of gel-based traditional foods such as tofu, yogurt, sausage and meatballs, not only improving their texture but also holding and protecting moisture, fat, flavoring substances and bioactive molecules [1], as well as controlling the release of the substances *in vivo* and *in vitro* [2]. 

Depending on gelation temperature, protein gel is divided into the heat- and cold-induced types. A high concentration of protein is required for heat-induced gel, and forms gel at above its denaturation temperature, but cold-set gel is quite different from the heat-induced one in terms of gelling conditions; it mainly involves a relatively low protein concentration and temperature, heat-denatured aggregates, is an inducer in most cases, and so on. Therefore, protein cold-set gels are more suitable for embedding and protecting heat-sensitive bioactive ingredients. Some proteins (including soybean protein isolate, porcine plasma protein, collagen, bovine serum albumin and whey protein isolate, etc.) have been reported to form cold-set gel [3,4]. However, protein-only-based cold-set gels are deficient in texture in most cases. Because of this, polysaccharides are often used to improve the performance of protein gel [5,6]. A large number of studies have found that polysaccharides can interact with proteins in aqueous solutions to promote the formation of the protein gels and improve the performance of the gels due to the increase in non-chemical bond forces and intermolecular disulfide bonds [5]. Starch, as one of the most widely used polysaccharides, often has certain positive effects on protein gels, and there are different opinions about the mechanism. In addition to the possible mechanisms mentioned above, one idea is that the ability of starch to absorb water plays a role in binding and stabilizing free water molecules, thereby reducing the free water content in its surrounding microenvironment; this indirectly increases the concentration of proteins in the coexisting system, resulting in an enhancement of the proteins gel properties. Another view is that starch particles act as active fillers to strengthen the gel network [7]. However, there is no universal theory applicable to all polysaccharides in this regard. In fact, different polysaccharides have very different effects on protein gels, and the specific mechanisms are also different.

Bovine serum albumin is a transport globulin containing 583 amino acid residues in bovine blood. BSA heat-induced gel is well-known, and its properties are closely related to its properties are closely related to pH, ion and other conditions of the system [8]. So far, there are few reports on BSA cold-set gels, despite the excellent ability of BSA to bind bioactive molecules, and thereby BSA cold-set gel has not been fully explored as a vehicle for molecular embedding, protection and delivery. The previous studies have found that heat-denatured BSA (hdBSA) aggregates could be induced to form cold-set gel in the presence of glucose-δ-lactone (GDL) and NaCl [9]. In addition, microbial transglutaminase (MTGase) and GDL could also synergistically induce the formation of BSA cold gel [10]. BSA cold-set gel could be a good carrier for bioactive molecules [9,10], but some deficiencies remained to be improved, including unsatisfactory texture, water-holding capacity (WHC) and controlled-released ability at lower BSA concentration.

Starch and protein could interact with each other during food processing, and thereby improve the texture, physicochemical properties and digestibility of the food. Some research indicated that the textural profiles of protein gels could be significantly changed by the addition of starch, and affected by types of protein, types of starch, and their concentrations and ratios [11,12]. For example, elasticity and hardness of whey protein gel could be increased through adding native potato starch [13]; 10% acetylated distarch phosphate enhanced hardness and chewiness of soybean thermal gels, while the addition of native starch resulted in a decrease in textural properties [12]; lotus root amylopectin could induce the formation of an uniform and compact whey protein isolate three-dimensional network gel structure, through increasing the sulfhydryl, C-N and N-H bonds in the gel [14]. On the other hand, some studies have shown that protein could inhibit starch gelatinization and retrogradation, probably due to its steric restriction on the recrystallization process of the starch [15,16,17], and thereby in turn improve the quality of starchy foods. Soluble starch (SS) is a derivative of starch, processed by oxidant, acid, enzyme and other methods, which can be dissolved into a transparent solution with low viscosity in boiling water, and which is more stable than the original starch and dextrin. SS has important uses in food and medicine, but no studies have addressed the effect of SS on protein gels.

In this work, in order to further improve the texture performance of BSA cold-set gel and enhance its controlled-release ability as a bioactive factor-embedding system, BSA-SS composite gel was investigated. The effect of SS on BSA cold-set gel was first explored, and then its mechanism was also probed through detecting the water state and the interaction between BSA and SS. Moreover, quercetin was applied as a model of a bioactive factor to research the controlled-released effect of the composite gel, according to the method of simulating gastrointestinal digestion. 

## 2. Materials and Methods

### 2.1. Materials

BSA (purity ≥ 98%) was purchased from Saiguo Biotechnology Co., Ltd. GDL was purchased from Sigma-Aldrich (St. Louis, MO, USA). SS (from potato, analytical purity) was supplied by Sinopharm Chemical Reagent (Shanghai, China). MTGase (containing 1.0 wt% of the enzyme) was purchased from Yiming Biological Products (Taixing, China). Quercetin (purity ≥ 97%) was from Macklin Biotechnology Co., Ltd. (Shanghai, China). 

### 2.2. Preparation of BSA-SS Cold-Set Gel

The SS stock solution was first prepared through heating SS dispersion for 1 h in a water bath of 85 °C with magnetic stirring, and it was then dropped into BSA solution (pH7.0) with magnetic stirring to obtain BSA-SS gradients containing SS of 0.0%, 1.0%, 2.0%, 3.0% and 4.0% (*w*/*v*), respectively, followed by heating for 20 min at 85 °C, in which BSA was 4.0% (*w*/*v*). Finally, the cooled BSA-SS complex solutions (named as BSA4-SS0, BSA4-SS1, BSA4-SS2, BSA4-SS3, and BSA4-SS4, respectively) were kept for later use.

The gel was fabricated by using GDL and MTGase as inducers [4]. GDL powder and MTGase solution was added to the above BSA-SS gradients, and reached up to the working concentration of 0.3% (*w*/*v*) and 0.2% (*w*/*v*), respectively; this was immediately followed by pouring 5 mL of the mixture into a petri dish and gelling for 4 h at room temperature. 

### 2.3. Characteristics of BSA-SS Cold Gel 

#### 2.3.1. Texture Determination

According to Zhang’s method [4], hardness and springiness of the gel were determined using a TA-XT plus texture analyzer (Stable Micro System, Godalming, UK), and P/0.5R cylindrical probe, and two compression measurement modes were used. The sample compression degree was 30%, with a trigger force of 5 g. 

#### 2.3.2. Rheological Analysis

A rheometer (MCR93, Anton Paar (Shanghai) Trading Co., Ltd., Graz, Austria) was applied to analyze the rheological characteristics of the gels [9]. The loss modulus (G″) and storage modulus (G′) were measured. The sample-loaded flat plate of 50 mm in diameter was covered with silicon oil to maintain a constant humidity. Time scanning was carried out for 4 h at 25 °C under a constant stress and frequency of 0.5 Pa and 1 Hz, followed by frequency scanning (0.1–10 Hz) at 0.5% strain. 

#### 2.3.3. Microstructure

The freshly prepared gels were immediately lyophilized after being made brittle in liquid nitrogen, and then their microstructures were observed using scanning electron microscopy (SEM) (JSM-9360LV, JEOL, Ltd., Tokyo, Japan), after creating a sputtered-gold coating. 

#### 2.3.4. Water-Holding Capacity (WHC)

The centrifugation method was use to evaluate WHC [18]. A total of 5 g or so of gel (*W*_1_) was centrifuged at 10,000 rpm for 15 min, and then blotted the separated water from the gel was blotted and weighed (*W*_2_). The formula was applied to calculate the WHC.
$$\mathrm{WHC}(\%)=\frac{W_{2}}{W_{1}}\times 100\%$$

#### 2.3.5. Determination of Water Binding State 

The sample was prepared according to the Reference [19]. The freshly prepared gel (0.5 × 1.0 × 1.0 cm) was placed into a nuclear magnetic tube of 1.5 cm in diameter. A low-field nuclear magnetic resonance (LF-NMR) analyzer (NMI20-015-1, Niumag Electric Corporation, Shanghai, China) was applied to analyze the state of water in the gels. 

### 2.4. Interaction between SS and BSA in the Gel

#### 2.4.1. Fourier-Transform Infrared Spectroscopy (FTIR)

A Nicolet iS 10 FTIR spectrometer (Thermo Fisher, Walsham, MA, USA) was used to record the FTIR spectra (4000–400 cm^−1^) of the lyophilized BSA-SS cold-set gel, which were recorded. A background spectrum in the air was obtained for each sample. 

#### 2.4.2. Zeta Potential

Zetasizer NanoZSP (Malvern, Worcestershire, UK) was used to measure zeta potential at 25 °C after appropriate dilution. 

#### 2.4.3. Fluorescence Spectrum Analysis

The cooled BSA-SS complex solutions were diluted to a BSA concentration of 0.05% (*w*/*v*) to determinate the fluorescence spectra using an F-7000 Fluorescence Spectrophotometer (Hitachi, Tokyo, Japan). BSA was excited at 280 nm to emit fluorescence.

#### 2.4.4. BSA Secondary Structure by Circular Dichroism (CD)

The cooled BSA-SS complex solutions prepared as described in Section 2.2 were diluted to a BSA concentration of 0.01% (*w*/*v*) with deionized water, and then the far ultraviolet circular dichroism spectra (190–240 nm) were determined at 25 °C using a J-1500 circular dichroism spectrometer (JASCO, Tokyo, Japan). Secondary structures of BSA were analyzed using specialized software for the instrument (Spectra Manger version 2.0, JASCO). 

### 2.5. Performance Evaluation of BSA-SS Cold-Set Gel as Carrier 

#### 2.5.1. Determination of Embedding Efficiency (EE) on Quercetin

Quercetin stock solution (0.4%, *w*/*v*) was prepared using absolute ethanol as solvent, and added into the cooled BSA-SS complex solutions; the quercetin in each sample was 0.02% (*w*/*v*), and this was followed by fabricating the gel.

The EE of BSA-SS cold-set gel was determined according to the Reference [20]. The whole gel was loaded in a tube to centrifuge at 10,000 rpm for 15 min, discarding the separated water. The gel was taken out and blotted the water off the surface with tissue paper, mixed with 20 mL of ethanol and subjected to ultrasonic treatment for 30 min and centrifuged at 8000 rpm for 10 min; then the supernatant was collected. The residue after centrifugation was extracted again under the same conditions. The supernatants of the two extractions were mixed and quantified. The formula was used to calculate EE, where *W*_1_ and *W*_0_ were the weight of the quercetin in the gel after and before being centrifuged, respectively.
$$\mathrm{EE}(\%)=\frac{W_{1}}{W_{0}}\times 100\%$$

#### 2.5.2. *In Vitro* Simulated Release of Quercetin

The release profile of quercetin from the gels was measured according to the Reference [10]. Four kinds of simulated digestive solutions were designed, namely HCl-saline solution (NSGF, pH 2.0, 0.1 M), phosphate-buffered saline (NSIF, pH 7.4, 0.1 M), simulated gastric fluid (SGF, HCl-saline solution with 0.1% pepsin, *w*/*v*) and simulated intestinal fluid (SIF, pH 7.4 PBS with 1.0% pancreatin, *w*/*v*). The quercetin-loaded gels were cut into small pieces of 5 × 5 × 5 mm (~1.0 g) using a scalpel, and then immersed in 20 mL of NSGF, SGF, NSIF and SIF, respectively, followed by digesting for 4.0 h on a shaker of 37 °C at 100 rpm. The dynamic release rate was determined by taking 2 mL of digestive solution at 30 min intervals. Quercetin contents in the digestive solutions were quantified according to the absorbance value at 374 nm after centrifuging at 10,000 rpm for 10 min. The release rate (RR) was obtained by the formula below, in which *C*_1_ and *C*_0_ (0.02%, *w*/*v*) were the content of quercetin in the digestive solution and in the original gel, respectively.
$$\mathrm{RR}(\%)=\frac{C_{1}}{C_{0}}\times 100$$

### 2.6. Statistical Analysis

All tests were repeated more than three times, and the results were expressed as means ± standard deviations. Significance of difference was analyzed using Duncan’s test in SPSS 21.0 (SPSS Inc., Chicago, IL, USA) and the level was set at *p* < 0.05.

## 3. Results and Discussion

### 3.1. Effect of SS on BSA Cold-Set Gel

#### 3.1.1. Texture Profile Analysis

Texture, including hardness, elasticity, chewability, etc., is the most important attribute of gel products, and starch is often applied to improve the texture of protein gel, due to the interaction between the two macromolecules [21]. Figure 1 shows that hardness and springiness of BSA cold-set gel induced by GDL and MTGase increased significantly with the increase in SS, especially when the concentration of SS reached or exceeded 3.0% (*w*/*v*). Under the same preparation conditions, BSA-free SS samples all showed typical liquid characteristics, according to our observations, so it could be inferred that the gelation of the BSA-SS composite system was based on the network structure formed by the cross-linking of BSA, and that the SS itself could not form gel, but only affected the gel texture by affecting the protein. 

Effects of starch on protein gel texture are quite different by different types of starch. In most cases, starch could only improve protein gels to a limited extent, and even the excessive addition of certain starches could destroy the formation of the protein gel network, resulting in the decrease in the gels’ strength [15]. It was an encouraging factor of the work that a relatively low concentration of gelatinized SS had a significantly positive influence on textural properties of BSA cold-set gel. Some research proved that texture improvement of protein gel was probably correlated with electrostatic repulsion between starch and the protein; swollen granules of the starch were dispersed in the continuous protein gel [13,22]. The Z-potential measurement showed that SS was almost uncharged and electrically neutral, while BSA carried a large amount of negative charge (Section 3.2.2) at the corresponding pH value. Therefore, there was no strong electrostatic repulsion between BSA and SS in the composite system. It could be inferred that SS did not fill the protein gel in the form of immiscible swelling particles to enhance the gel texture. The specific reasons for the improvement in texture need to be further supported by the data in the following sections.

#### 3.1.2. Changes in Rheological Characteristics during Gelation

There is a positive correlation between the gel texture and its rheological parameters of loss modulus (G″) and storage modulus (G′); that is to say, a larger value of G′ implies a stronger and correspondingly harder gel, and a smaller ratio of G′ to G″ means a more typical solid-like feature of the gel. Figure 2A shows BSA cold-set gelation profiles by adding SS of the different concentrations (1.0–4.0%). The G″ and G′ of all the samples remained constant shortly under the combined action of GDL (0.3%, *w*/*v*) and MTGase (0.2%, *w*/*v*), and then increased dramatically, to reach an approximate plateau within 2 h. It was worth noting that the G′ values of each BSA-SS composite system after gelling were much higher than that of the BSA-only system, in which the Gʹ of BSA4-SS4 was approximately 5 times as high as that of BSA4-SS0, suggesting some interactions between hdBSA and gelatinized SS during the gel formation process. 

A gelation time (t_g_), defined as the time when G′ equals to G″, was applied to evaluate the kinetics of gelation. Obviously, SS accelerated the gelation and shortened t_g_. The t_g_ for the BSA-only system was 41 min, and then decreased markedly, along with the increase in SS, reaching up to approximately 9 min for BSA4-SS4 (Figure 2B). Some other polysaccharides also had a similar tendency in affecting protein cold gels, not only increasing the viscosity and elasticity of the gels, but also shortening t_g_ [23,24]. It is well known that the common factors affecting protein gel strength and t_g_ involve protein concentration, pH, certain ions, crosslinking agents and so on, among which the strengthening and accelerating of the protein cold-set gel resulting from increasing protein concentration could be related to the formation of some larger aggregates and the increase in protein-connected regions in the gel network [25,26]. Here, except for the increase in SS concentration, the other factors did not change, and it was demonstrated by our experiments that SS alone could not form a gel, even at a concentration of 4%. However, starch was reported to be able to increase the local concentration of the protein in a system of protein–starch coexistence, due to water expansion of the starch particles [27]. Therefore, the changes in the gelcaused by increased SS were likely associated with the constraints of the SS granules on free water, which is further discussed in Section 3.1.4. 

The frequency dependency characteristics of the samples with different concentrations of SS in the presence of 0.2% (*w*/*v*) MTGase and 0.3% (*w*/*v*) GDL are presented in Figure 2C. All samples had no obvious frequency dependence in the low-frequency region (0.1–10 Hz), suggesting a typical continuous network [28].

#### 3.1.3. Microstructure of the Cold-Set Gels

In order to understand the effect of SS on the mechanical properties of BSA cold-set gel, the microstructures of the gels were observed (Figure 3). A total of 4.0% (*w*/*v*) of the gelatinized SS could not form a gel, and the lyophilized SS showed the stack of fibrous structures. BSA-only cold-set gel was formed by a typical three-dimensional network with a thin wall; the pore sizes in the gels were gradually reduced with the increase in SS and the compactness of the gel increased gradually, which corresponded to the increase in gel strength. 

#### 3.1.4. Change in WHC and Water Binding State

WHC can reflect the ability of gel to fix and capture water [29], as is shown in Figure 4A. The WHC of BSA-only (4.0%, *w*/*v*) cold-set gel was 48.54%, which was likely due to the poor gel network caused by a low concentration of BSA. However, WHC increased by 13.33%, 32.44%, 52.36% and 57.19%, along with increase in SS, respectively, and there was no significant difference between BSA4-SS3 and BSA4-SS4. The results were consistent with the texture and rheological characteristics of the gel. Many studies have shown that polysaccharides not only made the network structure of the protein gel more compact, but also bonded more water through hydrogen, thereby increasing the WHC [14], which is highly consistent with our results. Therefore, the results for the WHC implied that the change in water state in the gel system with the increase in SS could be the key factor affecting the gel properties.

LF-NMR has been widely used as an analytical method to determine the binding state and distribution of water in the gels. It can be seen from Figure 4B and Table 1 that the T_2_ spectra had three peaks distributed within the relaxation time of 1–10,000 ms, described as T_21_, T_22_ and T_23_, representing binding water, fixed water and free water, respectively [30]. T_21_, T_22_ and T_23_ shifted gradually in the direction of low relaxation time with the increase in SS, suggesting the better water-restriction capacity of the BSA-SS composite cold-set gel compared to the BSA-only one, and the higher the starch content, the stronger the capacity. Furthermore, PT_23_ significantly decreased from 12.05% (BSA4-SS0) to 3.37% (BSA4-SS4), and PT_22_ increased from 89.65% (BSA4-SS0) to 96.26% (BSA4-SS4) (*p* < 0.05), but there was no significant difference in the proportion of PT_21_ (*p* > 0.05), suggesting that some free water converted to immobilized water, due to the addition of SS. Some researchers believe that the enhancement of protein gel caused by starch is related to the removal of water from the system by starch particles [26]; that is to say, due to the coexistence of starch, part of the moisture in the protein gel system is combined with the starch, thereby reducing the moisture participating in the protein gelation and increasing the protein concentration in the local microenvironment. This is directly related to the gel performance and velocity. Therefore, it could be inferred that SS bound much of the original free water in the BSA gel, thereby indirectly increasing the concentration of BSA and improving the texture properties of the gel and speeding up the gelation.

### 3.2. Acceleration Mechanism and Performance Improvement of BSA Cold-Set Gelation by SS

#### 3.2.1. FTIR Spectroscopy Analysis 

Figure 5 showed the FTIR spectra of the freeze-dried samples. The main characteristic absorption bands, amide I and amide II, of the BSA-only (BSA4-SS0) resulting from peptide linkages, peaked at 1644.5 cm^−1^ and 1531.2 cm^−1^. The fingerprint area of starch at 800–1200 cm^−1^ reflected the ratio of ordered and amorphous structures in starch [31]. The addition of SS caused a peak shift in the amide I band from 1644.5 cm^−1^ to 1651.3 cm^−1^, showing a trend toward a higher wave number, which suggested some changes in the protein backbone structure through the role of starch [32]. Furthermore, a wider shoulder peak of the amide I band was also observed in BSA-SS systems, probably attributed to the formation of hydrogen bonds and the elaboration of protein internal functional groups [33]. The peak of the amide II band shifted gradually from 1531.2 cm^−1^ (BSA-only) to 1536.5 cm^−1^ (BSA4-SS4). In the BSA-SS composite system, the absorption band of SS between 1680 cm^−1^ and 1795 cm^−1^ also disappeared. 

The bands between 3100 and 3700 cm^−1^ were designated as O-H stretching, related to the intra- and intermolecular hydrogen bonds, and the peak of BSA-SS cold-set gel shifted from 3300 cm^−1^ (SS only) to 3280 cm^−1^; this was probably due to a significant increase in hydrogen bond density and strength through the decrease in molecular mobility in the starch–protein mixtures [11,34]. No new peaks were generated, and thus it could be speculated that only non-covalent interactions occurred during BSA-SS cold-set gelation. Some researchers attributed the dense and stable protein gel strengthened by the starch to the cross-linking of proteins and starch [14], while other reports have shown that starch/flour and protein interact through some secondary bonding forces, to improve the protein gel [35,36,37].In the BSA-SS composite gel, no new chemical bonds were caused; instead, more and more hydrogen bonds were formed, along with the increase in SS, which could have resulted from the interaction between SS and BSA. Moreover, starch as a physical filler in different forms in the voids of the gel has also been reported to be responsible for the strengthening of the gel [38]. Therefore, the enhancement of BSA cold-set gels by SS could originate from the interaction between BSA and SS and the increase in BSA concentration, due to the effect of SS on water removal; the physical filling effect of SS in the gel could also contribute a little to the effect.

#### 3.2.2. Zeta Potential of BSA-SS System

Starch is generally considered as a neutral macromolecule. Figure 6A showed that BSA in the absence of SS possessed many net negative charges at neutral pH, and the zeta potential was about −30 mV. The zeta potential of SS was determined to be about −1 mV to −2 mV under the same conditions, but the negative charges of the BSA-SS system decreased with the increasing SS. Similar results were also found in whey-protein-isolate-potato starch, due to an increasingly strong interaction between the two, with the increase in the starch [39]. It could be speculated that it was almost impossible for BSA to bind to SS and weaken its electronegativity through electrostatic interaction. Therefore, the zeta potential reduction in the BSA-SS system could have resulted from the spatial shielding effect of the SS macromolecules on the negative charges on the surface of BSA, through binding to the BSA by hydrogen bonds. The researchers believed that the colloidal particles in the protein-polysaccharide mixed system with minimized charges could be more rapidly aggregated [40], and thereby provide the appropriate precursor more rapidly, to facilitate the gelation reaction by GDL and MTGase [4]. 

#### 3.2.3. Fluorescence Quenching Analysis

Fluorescence quenching is an effective method to probe into the interaction between the protein and the other molecule [41]. The intrinsic fluorescence emission profiles of hdBSA with various concentrations of SS were determined, to further explore the forming of BSA-SS complexes (Figure 6B). The hdBSA exhibited a strong fluorescence emission peak at 340 nm, and the peak decreased regularly with the increase in SS concentration, indicating that the addition of SS could obviously weaken the intrinsic fluorescence of BSA. The degree of fluorescence quenching was positively correlated with the SS concentration, as this could be attributed to the adsorption of SS to BSA, to form BSA-SS complexes [42,43]. It was found that aromatic side chains of protein such as tryptophan (Trp), tyrosine (Tyr), or phenylalanine (Phe) could interact with starch by non-covalent binding to form a stable complex, and thereby weaken the intensity of protein fluorescence [42,44]. Accordingly, the above results consistently indicated that BSA and SS formed the complexes at a neutral pH, by their interaction, which was consistent with the FTIR results.

#### 3.2.4. Circular Dichroism Analysis

CD spectrum scanning was carried out to explore the secondary structure of BSA with different concentrations of SS, and the results are presented in Figure 6C and Table 2. The CD spectrum of BSA possessed a negative peak at 208 nm and a positive peak near 192 nm, representing the α-helix in the protein. As the concentration of SS increased, there were no significant changes in the intensity and position of the peaks, and the data in Table 2 also indicated that secondary structures had not significantly changed. The results could suggest that interactions between BSA and SS could occur on the surface of the protein molecules or colloidal particles [38], and that SS had no obvious influence on the secondary structure of hdBSA. 

### 3.3. Embedding Ability of BSA-SS Cold-Set Gel

Quercetin is a plant flavonoid with chemopreventive and therapeutic effects on a variety of diseases, including antioxidant, anti-cancer, anti-hypertensive, anti-diabetic, anti-inflammatory effects, and improving immunity [45]. However, quercetin is poorly soluble in water and has a short half-life, and therefore has low bioavailability. Some studies showed that BSA and quercetin could form a complex and improve its solubility, stability and bioavailability [46]. In this work, quercetin was used as a representative of heat-sensitive active substances to preliminarily evaluate the embedding potential of BSA-SS composite cold-set gel.

The EE of quercetin in the cold-set gel was positively correlated with the SS concentration (in Figure 7), which increased from 68.3% to 87.45% (*w*/*v*) within the designed concentration range of SS. The previous research revealed that the number of adsorption sites in the microstructure of BSA cold-set gel could be directly related to its EE [9]. In addition, starch was likely to interact with quercetin by forming hydrogen bonds between the C-4’ and C-5’ hydroxyl groups of quercetin and starch, thereby improving EE [20,47]. Meanwhile, the spatial retention on quercetin in the gel network cannot be ignored, as it was directly related to the WHC of the gel.

### 3.4. In Vitro Release of Quercetin in BSA-SS Composite Cold-Set Gel 

Various gels and emulsions have a certain controlled-release effect, which varies greatly, due to the composition and structure of the system [48,49,50]. The addition of polysaccharides usually enhances the controlled-release effect of these systems. The quercetin-loaded BSA-SS composite cold-set gels with different SS concentrations (0.0%, 2.0% and 4.0%, *w*/*v*) were applied to investigate the controlled-released effect by an *in vitro* simulated digestion experiment, and the results of the release dynamics in SGF and SIF are shown in Figure 8. In the absence of digestive enzymes, the release rate of quercetin from NSGF was higher than that from NSIF, indicating that the gel network was pH-responsive [51]. The release rates of quercetin in the systems with digestive enzymes were obviously much higher than those of enzyme-free systems. The RR of quercetin in the gel of the SS free reached 85.13% and 37.24% after digesting for 4 h in SGF and SIF, respectively. The obvious decreasing trends of RR with the increase in SS were observed in the four digestive systems, suggesting that SS had a positive effect on the controlled release of the gel. The RR of quercetin was 10.42% lower in BSA4-SS4 than in BSA4-SS2 after digesting for 4 h in SGF, while it was only 3.26% in SIF. It was reported that a stable structure has good controlled-release effect, and so the addition of polysaccharides to protein gel could decrease the susceptibility of gels to enzymolysis, due to the interaction between the protein and polysaccharide, resulting in a strong network [50,52]. The results of controlled release also suggested that SS effectively enhanced the texture of the gel and played an important role in stabilizing the gel. Therefore, SS could be used as a controlled-released material to protect and improve the bioavailability of bioactive substances like the other starches, including potato starch and lotus root starch [20,53]. 

## 4. Conclusions

In this work, the effect and mechanism of SS on BSA cold-set gel was studied. The mechanical properties of BSA cold-set gel were significantly strengthened by adding pregelatinized SS, and the gelation rate was greatly accelerated. The SS removed some free water in the system by binding, according to the results of LF-NMR and WHC, and thereby improved the composite gel through increasing indirectly the local concentration of BSA. Moreover, the interaction between BSA and SS could play a role in strengthening the gel, and the hydrogen bond could be the main binding force between the BSA and SS in the gel. The SS reduced the negative charges of BSA aggregates by binding to the BSA, and thereby promoted the rapid formation of the gel aggregates; this accordingly accelerated the gelation process induced by the GDL and MTGase. The BSA-SS composite cold-set gel could be a potential controlled-released carrier to load and protect some heat-sensitive bioactive substances. The work provided a novel protein–polysaccharide composite cold-set gel with excellent performance, and provides a strategy for the design of a hydrophilic colloid carrier in the functional food, pharmaceutical and cosmetic fields.

## Figures and Tables

**Figure 1 foods-12-04313-f001:**
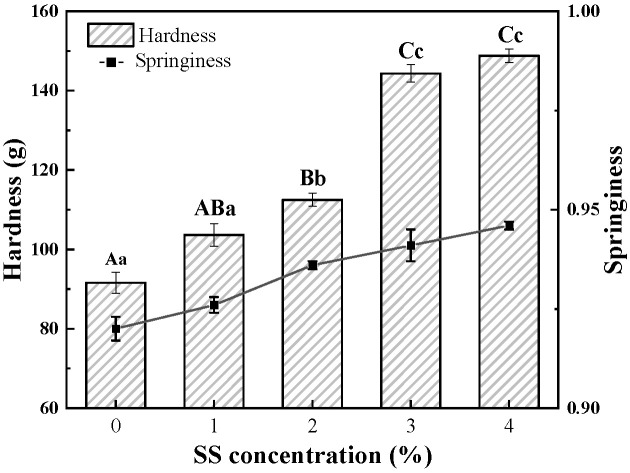
Texture of BSA-SS cold-set gels with different concentrations of SS. Data points represent means (n = 3) ± standard deviations (SD). Different letters indicate the significance of difference in hardness (capital letters) and springiness (lowercase letters) among the samples (*p* < 0.05).

**Figure 2 foods-12-04313-f002:**
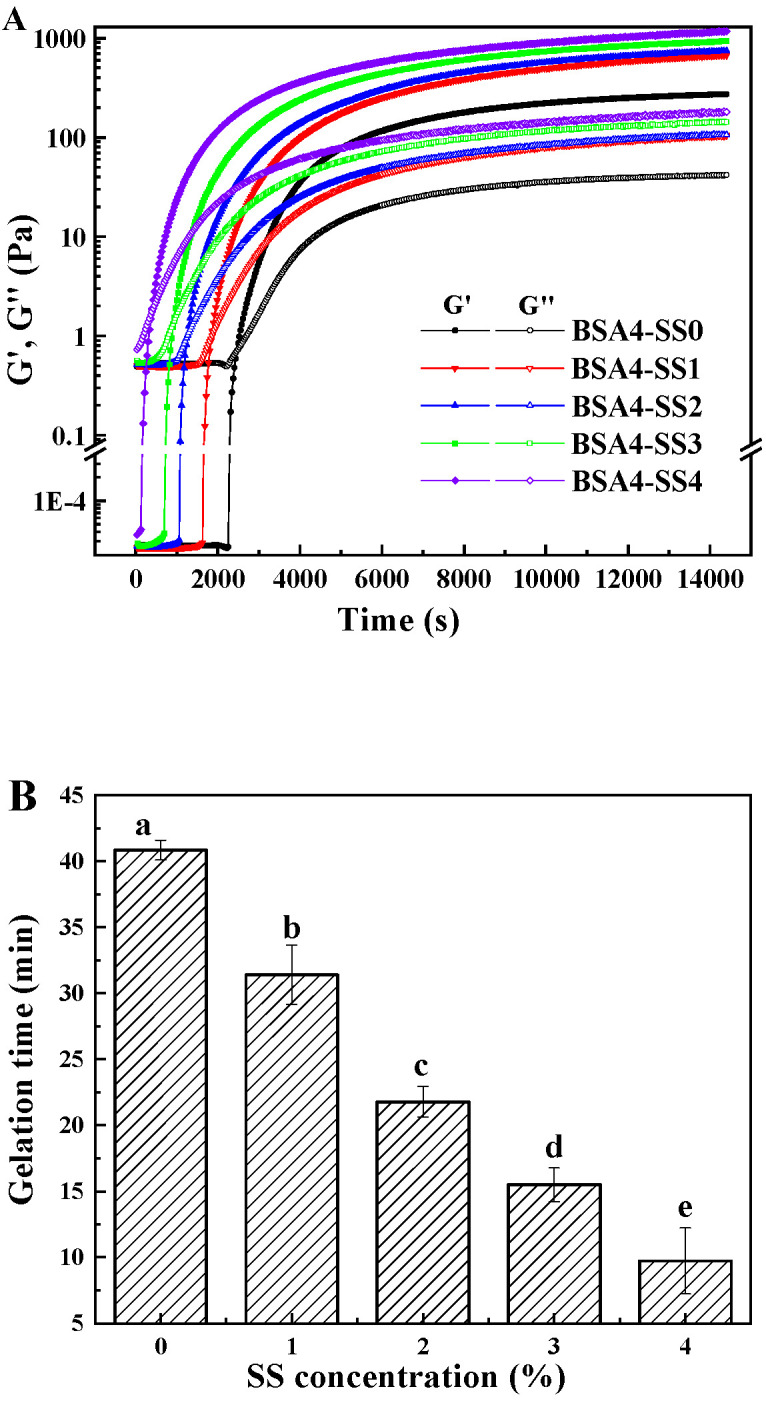
Rheological characteristics of BSA-SS composite cold-set gelation process. (**A**) Time scanning; (**B**) Time of gelling point (G′ = G″); (**C**) Frequency scanning. Storage modulus (G′, solid), and loss modulus (G″, open). SS concentrations were 0, 1.0%, 2.0%, 3.0% and 4.0% (*w*/*v*), respectively, and BSA concentration in each sample was 4.0% (*w*/*v*). Different letters (in (**B**)) indicate the significance of difference among the samples (*p* < 0.05).

**Figure 3 foods-12-04313-f003:**
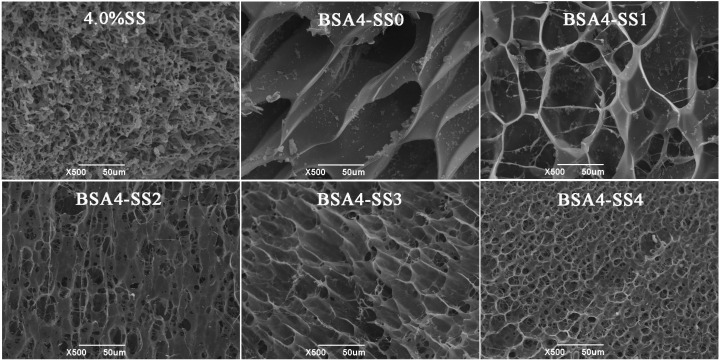
Microstructure of BSA-SS composite cold-set gels using SEM. The sample “4.0% SS” represents the gelatinized starch solution of 4% (*w*/*v*) after lyophilization, and “BSA4-SS0-4” means the freeze-dried gels with SS concentrations of 0, 1.0%, 2.0%, 3.0% and 4.0%, respectively; the BSA concentration of each sample was 4.0% (*w*/*v*).

**Figure 4 foods-12-04313-f004:**
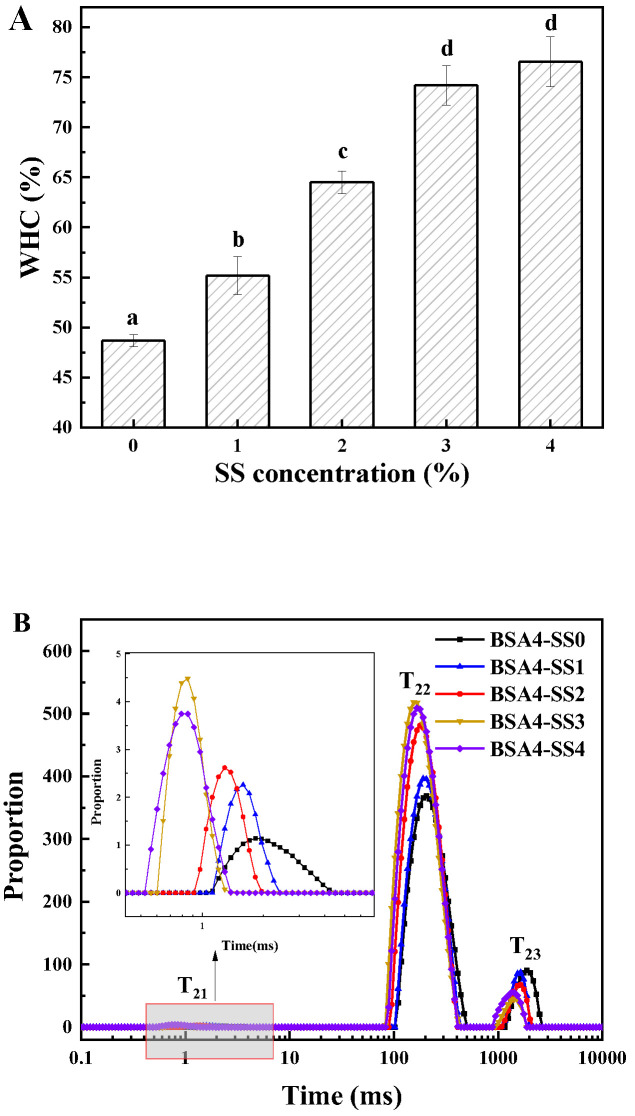
(**A**) Water-holding capacity (WHC) and (**B**) T_2_ relaxation-time distribution of BSA-SS composite cold-set gels with different concentrations of SS. BSA concentration in each sample was 4.0% (*w*/*v*). Different letters indicate the significance of difference among the samples (*p* < 0.05).

**Figure 5 foods-12-04313-f005:**
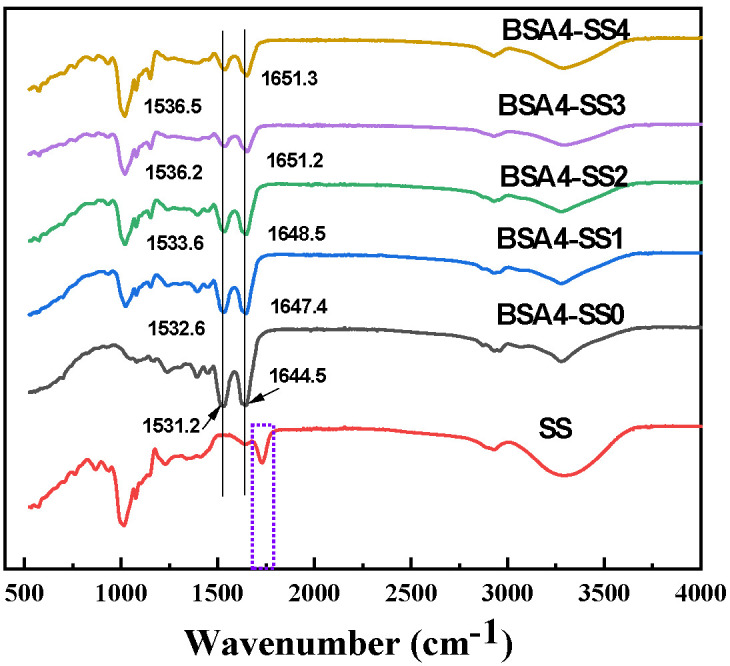
FTIR spectra of the lyophilized BSA-SS composite cold-set gels. The sample “SS” represents the gelatinized starch solution of 4% (*w*/*v*) after lyophilization, and “BSA4-SS0-4” means the freeze-dried gels with SS concentrations of 0, 1.0%, 2.0%, 3.0% and 4.0%, respectively. The BSA concentration in each sample was 4.0% (*w*/*v*).

**Figure 6 foods-12-04313-f006:**
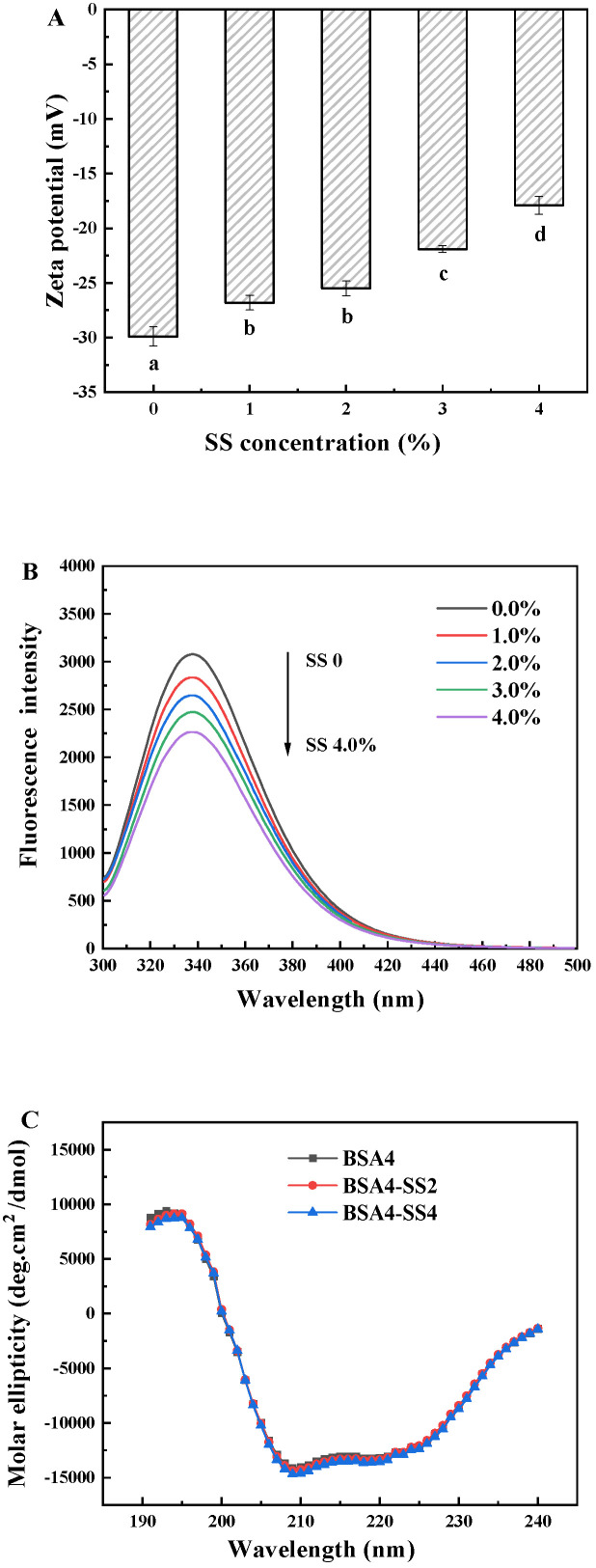
(**A**) Zeta potential, (**B**) fluorescence spectra and (**C**) circular dichroism spectra of the diluted BSA-SS aggregates containing different concentrations of SS and the same concentration of BSA. Data points represent means (n = 3) ± standard deviations. Different letters indicate the significance of difference among the samples (*p* < 0.05).

**Figure 7 foods-12-04313-f007:**
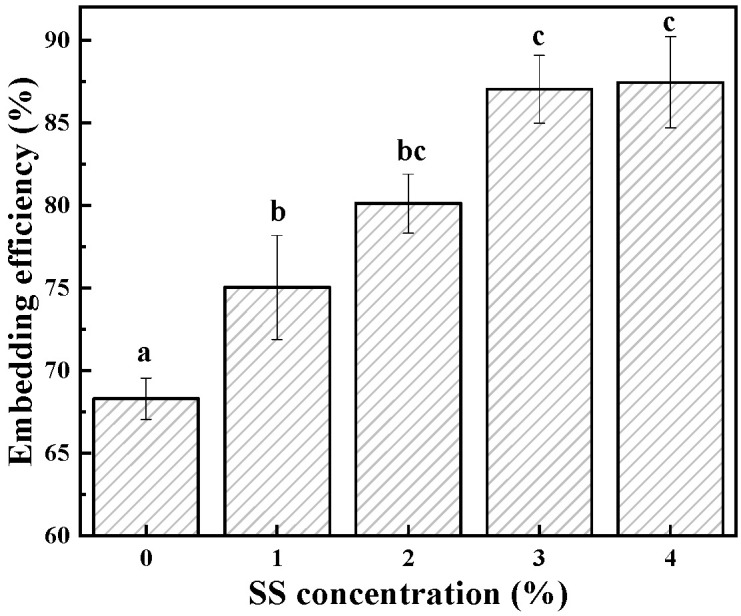
Embedding efficiency of BSA-SS composite cold-set gels with different concentrations of SS. Data points represent means (n = 3) ± standard deviations. Different letters indicate the significance of difference among the samples (*p* < 0.05).

**Figure 8 foods-12-04313-f008:**
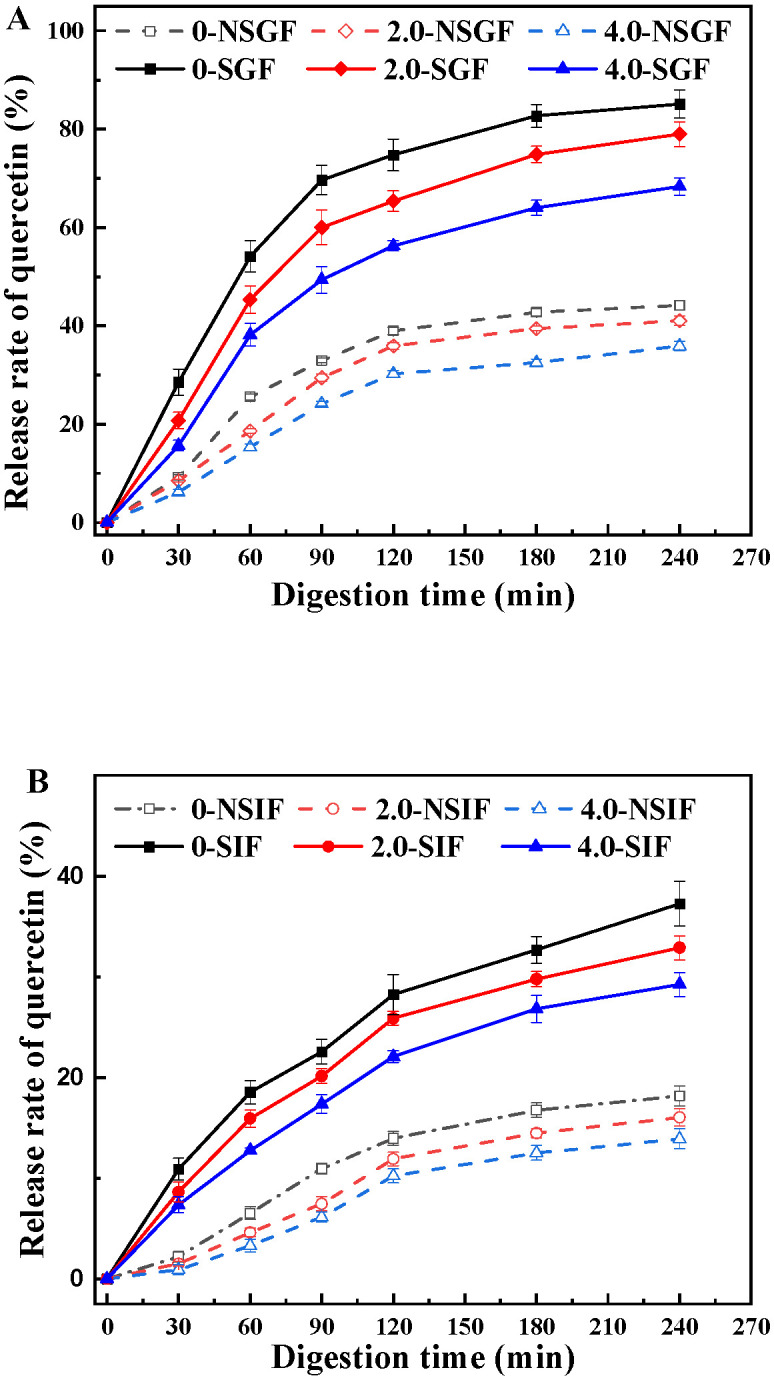
Release profiles of quercetin from BSA-SS composite cold-set gels. (**A**) in simulated gastric fluid (SGF, pH 1.2); (**B**) in simulated intestinal fluid (SIF, pH 7.4). The numbers (0, 2.0, 4.0) represent SS concentration in the gels; for example, 2.0-NSGF means the gel containing 2.0% (*w*/*v*) SS was treated with simulated gastric fluid without pepsin. Data points represent means (n = 3) ± standard deviations.

**Table 1 foods-12-04313-t001:** T_2_ relaxation times and relative peak areas of BSA-SS cold-set gels.

Sample	T_21_ (ms)	T_22_ (ms)	T_23_ (ms)	PT_21_ (%)	PT_22_ (%)	PT_23_ (%)
0	1.82 ± 0.03 ^d^	204.94 ± 0.14 ^d^	1889.65 ± 1.73 ^d^	0.30 ± 0.03 ^a^	87.65 ± 0.17 ^a^	12.05 ± 0.10 ^a^
1.0%	1.58 ± 0.04 ^c^	191.21 ± 2.25 ^c^	1644.67 ± 9.84 ^c^	0.34 ± 0.01 ^a^	90.15 ± 0.11 ^b^	9.51 ± 0.05 ^b^
2.0%	1.28 ± 0.07 ^b^	178.34 ± 3.91 ^b^	1534.36 ± 11.73 ^b^	0.36 ± 0.05 ^a^	93.35 ± 0.24 ^c^	6.29 ± 0.14 ^c^
3.0%	0.79 ± 0.14 ^a^	155.25 ± 4.73 ^a^	1431.45 ± 6.73 ^a^	0.38 ± 0.11 ^a^	96.14 ± 0.32 ^d^	3.48 ± 0.07 ^d^
4.0%	0.85 ± 0.08 ^a^	160.79 ± 4.21 ^a^	1445.14 ± 5.75 ^a^	0.37 ± 0.04 ^a^	96.26 ± 0.19 ^d^	3.37± 0.07 ^d^

Different letters (from “a” to “d”) represent the significance of difference between the samples (*p* < 0.05).

**Table 2 foods-12-04313-t002:** Secondary structure of hdBSA under different concentrations of SS.

Sample	α-Helix	β-Sheet	β-Turn	Random Coil
0	27.17 ± 0.07 ^a^	18.03 ± 0.09 ^a^	21.03 ± 0.18 ^a^	33.77 ± 0.05 ^a^
2.0%	27.00 ± 0.40 ^a^	18.13 ± 0.46 ^a^	21.11 ± 0.60 ^a^	33.76 ± 0.56 ^a^
4.0%	26.73 ± 0.36 ^a^	18.30 ± 0.27 ^a^	21.23 ± 0.76 ^a^	33.74 ± 0.46 ^a^

The same letter “a” represents no significant difference between the samples (*p* < 0.05).

## Data Availability

All the data presented in this study are available within the article.

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
