# Peer review of "Effect and Mechanism of Soluble Starch on Bovine Serum Albumin Cold-Set Gel Induced by Microbial Transglutaminase: A Significantly Improved Carrier for Active Substances"

_foods, 2023, doi:10.3390/foods12234313_

Round 1

Reviewer 1 Report

Comments and Suggestions for Authors

1. The main remark on the work is the lack of integrity and main positions of the article: goals and objectives, the main focus of research.

2. The introduction should clearly define the focus of the article: the article is devoted to the effect and mechanism of influence of soluble starch on cold bovine serum albumin gel. In this regard, provide known literature data on this problem using the example of mixtures of proteins and polysaccharides of different nature. In the introduction, the authors do not discuss the mechanism of action of polysaccharides on protein gels at all.

3. Based on the literature data presented by the authors, it is necessary to justify the purpose and objectives of this study. The article does not contain a statement of the purpose and task of the study. The phrase “In this work, the effect of soluble starch (SS) as gel modifier on BSA cold-set gel were investigated, and the interaction between hdBSA and gelatinized SS was also explored. Moreover, the control-released effect of the composite gel on bioactive molecule was also researched by the method of simulating gastrointestinal digestion and using quercetin as a model of hydrophobic bioactive substances. The work could provide a novel protein-polysaccharide composite cold-set gel with excellent performance, and a strategy for the design of hydrophilic colloid carrier in functional food, pharmaceutical and cosmetic fields” should be moved to conclusions, because it carries information about the work done.

4. The text of the article provides several options for the mechanism of the influence of starch additives on protein gelation under the proposed conditions:

• “the enhancement of protein gel caused by starch was related to the removal of water from the system by starch particles” (page 6);

• “the increase of protein concentration could lead to the strengthening and accelerating of its cold-set gel due to the formation of some larger aggregates and the increase of protein-connected regions in the gel network” (page 6);

• “Some researchers attributed the dense and stable protein gel by starch to the cross-linking of proteins and starch” (page 7);

• “very common that polysaccharides not only made the network structure of protein gel more compact, but also bonded more water through hydrogen, thereby increasing the WHC” (page 7);

• “suggested that some free water conversed to immobilized water due to the addition of SS” (page 8);

• “drawn that the interaction between BSA and SS in the gels could involve hydrogen bonds” (page 9);

• “the above results consistently indicated that BSA and SS formed the complexes at neutral pH by their interaction” (page 11);

The title of the article declares the mechanism of interaction between starch and protein. It follows from this that it should be proposed and justified by the authors, and not by reference to the works of other authors. It would be good in the article to propose and outline schemes for the interaction of protein and starch.

5. Conclusions should be redone. Conclusions to the work must correspond to the achieved goal and objectives, i.e. contain research results and the corresponding mechanism of interaction between starch and protein.

6. I would like to see an explanation from the authors: why the moisture absorption of the gel was not determined using the well-known ASTM D 570-98 method.

Author Response

Dear Reviewer:

Thank you very much for your comments on our manuscript entitled “Effect and mechanism of soluble starch on bovine serum albumin cold-set gel induced by microbial transglutaminase: A significantly improved carrier for active substances” (Manuscript Number: foods-2706148). The comments are all valuable and very helpful for revising and improving our manuscript, as well as important guiding significance to our research. We have studied the comments carefully and made correction which we hope to meet with approval. All proposed modifications and the added references are marked red in the revised manuscript.

The main corrections in the revised manuscript and the responses to Reviewer’s comments are as follows (the red part).

  1. The main remark on the work is the lack of integrity and main positions of the article: goals and objectives, the main focus of research.

Response: we agree with the comment. In the revision, we pay attention to improving the integrity and focus.

  1. The introduction should clearly define the focus of the article: the article is devoted to the effect and mechanism of influence of soluble starch on cold bovine serum albumin gel. In this regard, provide known literature data on this problem using the example of mixtures of proteins and polysaccharides of different nature. In the introduction, the authors do not discuss the mechanism of action of polysaccharides on protein gels at all.

Response: We have added the discussion about the mechanism, please see the second paragraph of the introduction in red.

  1. Based on the literature data presented by the authors, it is necessary to justify the purpose and objectives of this study. The article does not contain a statement of the purpose and task of the study. The phrase “In this work, the effect of soluble starch (SS) as gel modifier on BSA cold-set gel were investigated, and the interaction between hdBSA and gelatinized SS was also explored. Moreover, the control-released effect of the composite gel on bioactive molecule was also researched by the method of simulating gastrointestinal digestion and using quercetin as a model of hydrophobic bioactive substances. The work could provide a novel protein-polysaccharide composite cold-set gel with excellent performance, and a strategy for the design of hydrophilic colloid carrier in functional food, pharmaceutical and cosmetic fields” should be moved to conclusions, because it carries information about the work done.

Response: We have rewritten the last paragraph of the introduction to justify the purpose and objectives of the study, and the last sentence “The work could provide a novel protein-polysaccharide composite cold-set gel with excellent performance, and a strategy for the design of hydrophilic colloid carrier in functional food, pharmaceutical and cosmetic fields” has moved to conclusion after modification.

  1. The text of the article provides several options for the mechanism of the influence of starch additives on protein gelation under the proposed conditions:
  • “the enhancement of protein gel caused by starch was related to the removal of water from the system by starch particles” (page 6);

Response: This statement is more suitable for 3.1.4, so we have moved this sentence to 3.1.4 and rewritten it.

  • “the increase of protein concentration could lead to the strengthening and accelerating of its cold-set gel due to the formation of some larger aggregates and the increase of protein-connected regions in the gel network” (page 6);

Response: We have added and rewritten the second paragraph of 3.1.2 (in red) so as to propose and justify our opinions according to our results.

  • “Some researchers attributed the dense and stable protein gel by starch to the cross-linking of proteins and starch” (page 7);

Response: It seem to be not able to get the information from the results of this section, we moved this sentence to 3.2.1 for a more convincing analysis and add and rewrite it.

  • “very common that polysaccharides not only made the network structure of protein gel more compact, but also bonded more water through hydrogen, thereby increasing the WHC” (page 7);

Response: To make the analysis more logical and persuasive, we have modified and supplemented the first paragraph in 3.1.4 (in red).

  • “suggested that some free water conversed to immobilized water due to the addition of SS” (page 8);

Response: To make the analysis more logical and persuasive, we have modified and supplemented the second paragraph in 3.1.4 (in red).

  • “drawn that the interaction between BSA and SS in the gels could involve hydrogen bonds” (page 9);

Response: To make the analysis more logical and persuasive, we have modified and supplemented the second paragraph in 3.2.1 (in red).

  • “the above results consistently indicated that BSA and SS formed the complexes at neutral pH by their interaction” (page 11);

Response: we have modified and supplemented the second paragraph in 3.2.3 (in red).

The title of the article declares the mechanism of interaction between starch and protein. It follows from this that it should be proposed and justified by the authors, and not by reference to the works of other authors. It would be good in the article to propose and outline schemes for the interaction of protein and starch.

Response: Thank you very much for your careful review. After carefully studying the above-mentioned content, we agree with you very much. We have added the relevant content about the mechanism and obtained a unanimous understanding.

  1. Conclusions should be redone. Conclusions to the work must correspond to the achieved goal and objectives, i.e. contain research results and the corresponding mechanism of interaction between starch and protein.

Response: We have modified the conclusion according to the valuable comment (in red)

  1. I would like to see an explanation from the authors: why the moisture absorption of the gel was not determined using the well-known ASTM D 570-98 method.

Response: We are so sorry for not paying attention to ASTM D 570-98 method before. Centrifugal method has been adopted in many references for WHC determination of gels, and LF-NMR is an accurate and effective instrument for the determination of water status. Therefore, according to our existing conditions, we adopted these two methods to detect water in gels. ASTM D 570-98 is another method for determining the water absorption of materials, we will try to understand and use this method in future studies if possible.

We tried our best to improve the manuscript and made some changes in the manuscript. The changes will not influence the contents and framework of the manuscript.

We appreciate for the Reviewer’s warm work earnestly, and hope that the correction will meet with approval.

Once again, thank you very much for your comments.

Best regards,

Jianglan Yuan, Haoting Shi, Changsheng Ding

Reviewer 2 Report

Comments and Suggestions for Authors
Review foods-2706148

The report contains some new information. However, the mechanistic discussion needs further improvement. The elaboration should be broad enough thus peers can find the merit for food scientists. After all, BSA itself is a model protein but not a routine food protein.

One of the mechanisms should be mentioned is that starch addition affects the molecular interactions. Thus, the authors are suggested to search database like Web of Science with starch addition (Title) AND molecular interactions (Title) to get related reference to improve the discussion.
​
For references applied for discussion, the search scope should not be limited to starch as some other reports used flour rather than pure starch. However, some mechanism applied can be applicable for this current report. Therefore, the authors are suggested to search database like Web of Science with flour (Title) AND protein electrostatic interaction (Title) to get related reference for further discussion.

Though it is understandable this current report focuses on starch and protein system. However, it should be noted that ions, especially calcium ions, are critical for this gelation process, which was not assayed thus not sure how much affected. It would be good to mention this thus further investigation is needed to get more solid evidence to support the final conclusion. Thus, the authors can search database like Web of Science with calcium ion (Title) AND protein isolate gelation method (Title) to get related reference for enhancing discussion.

When protein crosslink is discussed, the improving effect on texture and rheological properties should be discussed together. Thus, the authors can search database like web of science with improving texture and rheological (Title) AND protein crosslink (Title) to get reference for further discussion. Similarly, database like Web of Science can be searched with promoted (Title) AND protein complex gel (Title) to get reference for mechanistic discussion.

For protein gelation, electrostatic interaction is important for protein-carbohydrate system. Thus, the authors are suggested to search database like Web of Science with protein gelation (Title) AND electrostatic interaction (Title) to get reference for discussion.

In addition, this electrostatic interaction is also closely related to gelation property. Thus, further search database like Web of Science with gelation property (Title) AND electrostatic interaction (Title) is needed to get reference for improving discussion.

​
The protein complex was characterized and it was claimed important for the release of active ingredients. Previous report mentioned that Pickering emulsion has similar effect. What is the difference among different systems? The authors can search database like Web of Science with characteristics protein complex (Title) AND Pickering emulsion (Title) to get reference to enhance the discussion.

The liquid gel system should be discussed combining structure and rheological property. Thus, the authors can search database like Web of Science with structure and rheological property (Title) AND liquid gel (Title) to improve the discussion.

The moisture content was tested. However, the discussion was in superficial. Acturally not just moisture content but also moisture state is critical, especially on microbiota composition of the food applied during storage. The authors can search database like Web of Science with moisture state (Title) AND microbiota composition (Title) for further discussion.

Normally active substance is related to metabolic pathway, with the term metabolomics. Thus, some discussion is needed on metabolic pathway related to in vitro release. For instance, the authors can search database like web of science with metabolic pathway (Title) AND in vitro release (Title) to get reference for discussion.

Correspondingly, in the introduction, some information related to food metabolomics should be mentioned related to food quality analysis. The authors can search database like Web of Science with food quality analysis (Title) AND metabolomics (Title) to get related reference to improve the section introduction.

Comments on the Quality of English Language

The overall quality of English is good. Some minor revisions are needed. 

Author Response

Dear Reviewer:

Thank you very much for your comments on our manuscript entitled “Effect and mechanism of soluble starch on bovine serum albumin cold-set gel induced by microbial transglutaminase: A significantly improved carrier for active substances” (Manuscript Number: foods-2706148). The comments are all valuable and very helpful for revising and improving our manuscript, as well as important guiding significance to our research. We have studied the comments carefully and made correction which we hope to meet with approval. All proposed modifications and the added references are marked red in the revised manuscript.

The main corrections in the revised manuscript and the responses to Reviewer’s comments are as follows (the red part).

The report contains some new information. However, the mechanistic discussion needs further improvement. The elaboration should be broad enough thus peers can find the merit for food scientists. After all, BSA itself is a model protein but not a routine food protein. 

 Response: We have strengthened the analysis and discussion of the enhancement mechanism of polysaccharide on protein cold-set gel in introduction and the results of relevant mechanisms (in red).

One of the mechanisms should be mentioned is that starch addition affects the molecular interactions. Thus, the authors are suggested to search database like Web of Science with starch addition (Title) AND molecular interactions (Title) to get related reference to improve the discussion.

​ Response: We have consulted the references in the database as suggested. It is believed generally that there is interaction between protein and starch in their composite gel, and the forces usually involve non-chemical bond forces. We have also supplemented the relevant analysis and references in the FTIR results and discussions (in red).

For references applied for discussion, the search scope should not be limited to starch as some other reports used flour rather than pure starch. However, some mechanism applied can be applicable for this current report. Therefore, the authors are suggested to search database like Web of Science with flour (Title) AND protein electrostatic interaction (Title) to get related reference for further discussion.

 Response: We have consulted the references in the database as suggested. A reference on flour with starch as its main component was selected to discuss in the FTIR results (in red) (Yamul and Lupano, 2005)

Though it is understandable this current report focuses on starch and protein system. However, it should be noted that ions, especially calcium ions, are critical for this gelation process, which was not assayed thus not sure how much affected. It would be good to mention this thus further investigation is needed to get more solid evidence to support the final conclusion. Thus, the authors can search database like Web of Science with calcium ion (Title) AND protein isolate gelation method (Title) to get related reference for enhancing discussion.

Response: It is true that calcium ions often have an important effect on protein or polysaccharide gels, but the two main reagents we used, BSA and SS, do not contain calcium ions, so we did not consider the effect of calcium ions on gels in this work.

When protein crosslink is discussed, the improving effect on texture and rheological properties should be discussed together. Thus, the authors can search database like web of science with improving texture and rheological (Title) AND protein crosslink (Title) to get reference for further discussion. Similarly, database like Web of Science can be searched with promoted (Title) AND protein complex gel (Title) to get reference for mechanistic discussion.

Response: we have modified the two parts according to the suggestion (in red in section 3.1.1 and 3.1.2)

 For protein gelation, electrostatic interaction is important for protein-carbohydrate system. Thus, the authors are suggested to search database like Web of Science with protein gelation (Title) AND electrostatic interaction (Title) to get reference for discussion.

 Response: It is true that electrostatic interactions often play an important role in protein-polysaccharide composite gel systems. The analysis of electrostatic interactions in our manuscript relies on the results of the section 3.2.2.  According to this result, the electrostatic interaction between BSA and SS in this system could be negligible. Moreover, according to the infrared analysis, the interaction force between BSA and SS in the gel is mainly hydrogen bond.

In addition, this electrostatic interaction is also closely related to gelation property. Thus, further search database like Web of Science with gelation property (Title) AND electrostatic interaction (Title) is needed to get reference for improving discussion.

​ Response: We believed that the interaction between BSA and SS is not significantly dependent on the electrostatic interaction in this gel system. The red characters in the revision are the supplement and improvement of the gel mechanism analysis, hoping to be more logical and persuasive.

The protein complex was characterized and it was claimed important for the release of active ingredients. Previous report mentioned that Pickering emulsion has similar effect. What is the difference among different systems? The authors can search database like Web of Science with characteristics protein complex (Title) AND Pickering emulsion (Title) to get reference to enhance the discussion.

Response: Gel and emulsion are two different systems. During controlled release, the emulsion usually seals the active substance in its oil droplets and releases it with the digestion of the oil-water interface layer during digestion, while the gel usually seals the active substance in its network structure and releases it with the disintegration of the matrix network structure. For more accurate analysis and comparison, we screened the literature on gel controlled release to strengthen the analysis and discussion in this part (in red in section 3.4).

 The liquid gel system should be discussed combining structure and rheological property. Thus, the authors can search database like Web of Science with structure and rheological property (Title) AND liquid gel (Title) to improve the discussion.

Response: When you say liquid gel, do you mean what the gel solution looks like without inducers? I can't find any literature on this. If I understand liquid gels correctly, we use liquid gels in some of our experiments to explore the mechanism of gel, including Zeta potential, Fluorescence quenching analysis, Circular dichroism analysis.

 The moisture content was tested. However, the discussion was in superficial. Acturally not just moisture content but also moisture state is critical, especially on microbiota composition of the food applied during storage. The authors can search database like Web of Science with moisture state (Title) AND microbiota composition (Title) for further discussion.

 Response: We have strengthened the analysis and discussion about the moisture, please see the red text in 3.1.4. As for the relationship between water and microorganisms, it is not relevant to the topic of this paper. Although moisture content and state have important impact on microbiota composition of the food applied during storage, I think it could be not appropriate to discuss because there is no data to support it in this manuscript. Nevertheless, we think this recommendation is very instructive because it suggests something we can focus on in future research.

Normally active substance is related to metabolic pathway, with the term metabolomics. Thus, some discussion is needed on metabolic pathway related to in vitro release. For instance, the authors can search database like web of science with metabolic pathway (Title) AND in vitro release (Title) to get reference for discussion.

 Response: In the work, we focused on the release of active substances in the gastrointestinal tract, and their metabolism is a bit off our subject. In addition, we believe that this release is not related to its metabolism in the body, because once the active substance is released and absorbed in intestine, its metabolism is no different from that of the free one. Moreover, I think it could be not appropriate to discuss this because there is no data to support it in this manuscript. Nevertheless, we think this recommendation is very instructive because it suggests something we can focus on in future research.

Correspondingly, in the introduction, some information related to food metabolomics should be mentioned related to food quality analysis. The authors can search database like Web of Science with food quality analysis (Title) AND metabolomics (Title) to get related reference to improve the section introduction.

Response: In the work, we focused on the release of active substances in the gastrointestinal tract, and their metabolism is a bit off our subject. In addition, we believe that this release is not related to its metabolism in the body, because once the active substance is released and absorbed in intestine, its metabolism is no different from that of the free one. Moreover, I think it could be not appropriate to discuss this because there is no data to support it in this manuscript. Nevertheless, we think this recommendation is very instructive because it suggests something we can focus on in future research.

Comments on the Quality of English Language

The overall quality of English is good. Some minor revisions are needed. 

Response: Thank you for the encouragement. We have modified some flaws in the language.

We tried our best to improve the manuscript and made some changes in the manuscript. The changes will not influence the contents and framework of the manuscript.

We appreciate for the Reviewer’s warm work earnestly, and hope that the correction will meet with approval.

Once again, thank you very much for your comments.

Best regards,

Jianglan Yuan, Haoting Shi, Changsheng Ding